

# Quenching a Fermi superfluid across the BEC-BCS crossover

Moritz Breyer, Daniel Eberz, Andreas Kell and Michael Köhl⋆

Physikalisches Institut, University of Bonn,
Wegelerstraße 8, 53115 Bonn, Germany

⋆ michael.koehl@uni-bonn.de

## Abstract

We study the response of a Fermi superfluid to a rapid change of the interaction strength. For a broad range of quench parameters, the order parameter exhibits damped oscillations, however, we also identify quench regimes in which these oscillations are absent. By comparing our data to a numerical model we find that the damping time constants are explainable by a dephasing in a local density approximation, however, the oscillation frequencies, while being comparable to the superconducting order parameter, display in detail significant differences to the integrable BCS (Bardeen-Cooper-Schrieffer) model.



## 1 Introduction

The response of superconductors to a sudden perturbation is of practical importance for devices such as superconducting detectors and switches. At the fundamental level, the physics of this process is driven by the highly interesting and non-trivial response of Cooper pairs to a change of their parameters, namely, whether they break up or undergo a quantum-coherent time evolution, and if so, for how long? The seminal contribution to our understanding of the dynamics of a superconductor following a sudden parameter change has been made by Volkov and Kogan [1]. Such a "sudden" perturbation requires a modification of the interaction

parameter within a duration short compared to the inverse Fermi energy, which, however, appeared out of reach at the time. In the limit of a small perturbation, they have found that the superconducting order parameter oscillates at twice the frequency corresponding to the superconducting gap, and these oscillations are damped with a power law. These oscillations of the order parameter have been identified with the Higgs quasiparticle mode [2–4].

After the experimental realization of ultracold gases in the crossover regime between Bardeen-Cooper-Schrieffer (BCS) superconductors and Bose-Einstein condensation (BEC) of dimers, the question of the dynamics gained renewed interest because the required short time scales for a "sudden" manipulation appeared to be within experimental reach. On the theoretical side, in particular, the integrable time-dependent BCS model [5] has provided theoretical insights including oscillation frequencies and damping. For example, for large quenches when the order parameter changes rapidly from an initial equilibrium value $\Delta_i$ to a significantly different final equilibrium value $\Delta_f$, the integrable BCS model predicts that the oscillations should occur at twice the value of the non-equilibrium order parameter [6–12], which, generally, is different from the endpoint of the quench $\Delta_f$. More recently, the complete theoretical phase diagram of the quenched superconductor has been determined in the integrable BCS model, which we show in Figure 1 for the homogeneous Fermi gas [8]. There are four distinct dynamical phases, labeled I, II, II′, and III, which correspond to distinctly different steady states. State I has a vanishing order parameter, states II and II′ exhibit a stationary order parameter after oscillations have attenuated in a power-law decay, and state III features a persistently oscillating order parameter. The difference between states II and II' can be understood by considering the two limits of the BEC-BCS crossover, which are separated by a sign change of the chemical potential. In the BEC regime II', the Higgs oscillation decays quickly into phonons due to a lack of particle-hole symmetry. As a result, the phases II and II' differ in the exponent of the power-law decay which is -1/2 (-3/2) in phase II (II') [13]. An extension to the harmonically trapped gas can, for example, be found in [9] which shows that the sharp phase boundaries soften.

Probing the response of a superconductor following an impulsive excitation has been realized both in BCS-type superconductors in the solid state [14, 15] and ultracold atomic gases [16, 17]. These measurements advantageously complement studies of the parametric excitation of the Higgs mode in these systems [18–20] because they directly look at the coherent dynamics of the order parameter rather than spectral features which could also include quasiparticle excitations. Cold atomic gases, in particular, allow for studying a broad parameter range of interaction strengths and amplitudes of the quench strength, which is unmatched in other experimental platforms, however, the quench phase diagram has not yet been explored.

## 2 Experimental realization

In this work, we study the dynamics of a strongly-interacting Fermi superfluid in the BEC-BCS crossover regime following a rapid quench of the interaction strength. We monitor the time evolution of the condensate fraction (as a proxy for the order parameter) following the quench and determine both frequency and damping constants. We access a large range of quench parameters within states II and II′ in the phase diagram (see Figure 1). States I and III are, unfortunately, out of our experimental range at present.

In our experimental work, we prepare an ultracold gas of $^6$Li atoms in a crossed-beam optical dipole trap with trapping frequencies $(\omega_x, \omega_y, \omega_z) = 2\pi \times (85, 137, 214)\,\text{Hz}$ in the harmonic approximation [21, 22]. The gas contains $\sim 4 \times 10^5$ atoms in a balanced mixture of the lowest and third-lowest hyperfine states at a temperature of $T \sim 0.1\,T_F$, well below the critical temperature. The interaction strength between the hyperfine states is controlled

**Sci**|**Post** SciPost Phys. 18, 053 (2025)

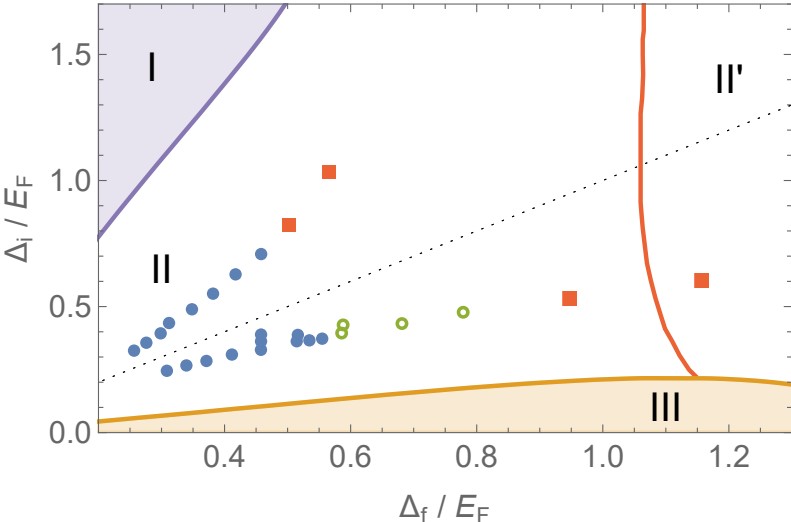

Figure 1: Theoretical phase diagram of the quenched Fermi superfluid (adapted from [8]) and experimental data obtained in this work. Solid circles denote the observation of decaying oscillations of the order parameter, open circles denote a time evolution explainable by dephasing oscillations within a local-density approximation (LDA), and squares are data for which the time evolution cannot be explained by dephasing oscillations within LDA. The dashed line corresponds to the small-quench scenario investigated by Volkov and Kogan [1], and the solid lines denote the phase boundaries between different steady states. State I has a vanishing order parameter, states II and II′ exhibit a stationary order parameter after oscillations have attenuated, and state III features a persistently oscillating order parameter. For more details, see main text.

with the help of a Feshbach resonance centered at 690 G [23]. After preparing the sample at a variable initial magnetic field, corresponding to an interaction parameter $1/k_F a_i$, we quench the interaction strength by changing the magnetic field by up to $\pm 36$ G within 2.7 μs. Here, $k_F = [3\pi^2 n(0)]^{1/3}$ denotes the Fermi wave vector and $n(0)$ is the density in the centre of the trap, which we determine by an inverse Abel transform from *in-situ* high-intensity absorption images [21, 22, 24]. Technically, the quench is performed with a custom-built magnetic field coil [25], which changes the magnetic field and therefore the *s*-wave scattering length $a$ from $a_i$ to $a_f$. After the quench, the system has a new interaction parameter $1/k_F a_f$. Since the quench is performed one order of magnitude faster than the Fermi time $h/E_F$ – in our case in the range of 12 μs to 25 μs – it is instantaneous for the system and the density remains unchanged. However, the order parameter immediately after the quench deviates from its new equilibrium value. We therefore realize a situation as studied in references [1, 7, 8, 10, 12, 13, 26–28]. This approach is comparable to pulsed THz excitations used to study the Higgs mode in superconductors [14, 29, 30]. We conduct a measurement of the dynamics of the order parameter after the quench by letting the system evolve for a variable hold time and then utilize the rapid ramp technique [31, 32] to measure the fraction of condensed atoms in our sample, which has a one-to-one correspondence to the order parameter. The rapid ramp is implemented by reducing the magnetic field to the zero-crossing of the scattering length (568 G for our spin mixture) within 700μs. Here, the field is stabilised for the expansion of the cloud. Before imaging, the field is ramped back close to the Feshbach resonance, where single atoms can be imaged. The condensed part of the cloud appears in the image as a non-Gaussian peak on top of a thermal background and the fraction of atoms in the non-Gaussian peak is determined by a two-dimensional fit.

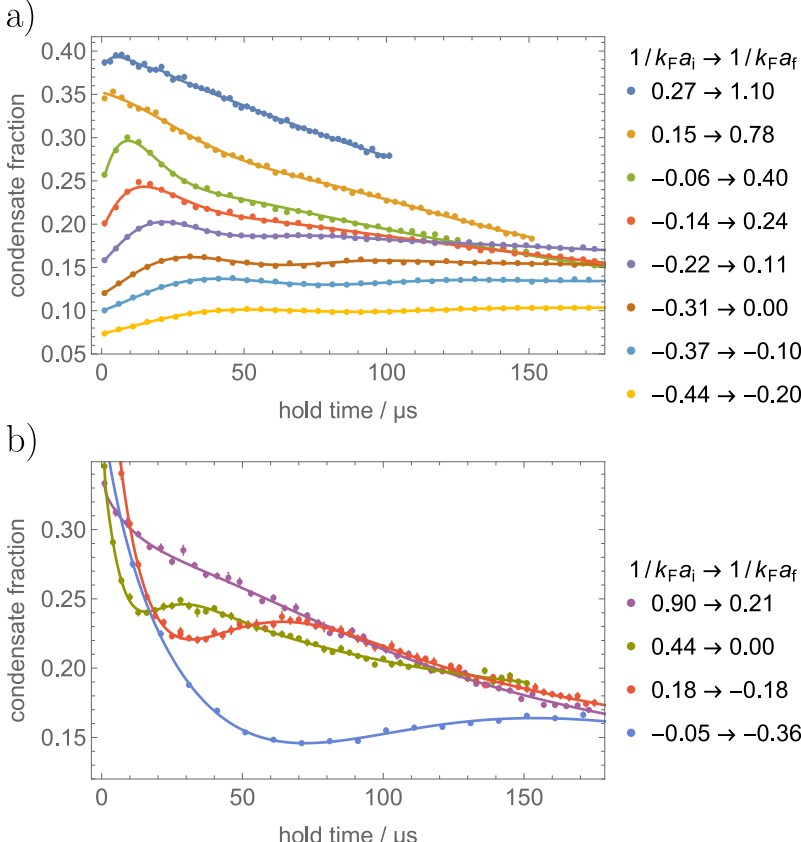

Figure 2: Experimental dynamics of the condensate fraction after a quench. a) Measured condensate fraction after a quench $1/k_{\mathrm{F}}a_{\mathrm{i}} < 1/k_{\mathrm{F}}a_{\mathrm{f}}$ for various interaction strengths. b) Measured condensate fraction after a quench $1/k_{\mathrm{F}}a_{\mathrm{i}} > 1/k_{\mathrm{F}}a_{\mathrm{f}}$ for various interaction strengths. The error bars denote the standard error of the mean, obtained after approximately 50 repetitions per data point. In a), the error bars are smaller than the symbols.

In Figure 2, we show samples of the time evolution of the condensate fraction for different combinations of initial and final values of the quench parameters. Qualitatively, we find different behaviour for different cases. Firstly, we study the case $1/k_{\mathrm{F}}a_{\mathrm{i}} < 1/k_{\mathrm{F}}a_{\mathrm{f}}$, see Figure 2a. For quenches starting and ending on the BCS side ($1/k_{\mathrm{F}}a_{\mathrm{i}} < 1/k_{\mathrm{F}}a_{\mathrm{f}} \lesssim 0$) we observe damped oscillations towards an equilibrium; for quenches from the BCS to the BEC side ($1/k_{\mathrm{F}}a_{\mathrm{i}} \lesssim 0 \lesssim 1/k_{\mathrm{F}}a_{\mathrm{f}}$), we observe a pronounced peak of the condensate fraction followed by a decay; for quenches well on the BEC side ($0 \lesssim 1/k_{\mathrm{F}}a_{\mathrm{i}} < 1/k_{\mathrm{F}}a_{\mathrm{f}}$), a decay of the condensate fraction. We also show the results of the opposite quench direction $1/k_{\mathrm{F}}a_{\mathrm{i}} > 1/k_{\mathrm{F}}a_{\mathrm{f}}$, see Figure 2b. For quenches on the BEC side, $1/k_{\mathrm{F}}a_{\mathrm{i}} > 1/k_{\mathrm{F}}a_{\mathrm{f}} \gtrsim 0$, we observe a simple decay of the condensate fraction and for quenches to the BCS side, $1/k_{\mathrm{F}}a_{\mathrm{i}} > 1/k_{\mathrm{F}}a_{\mathrm{f}}$ with $1/k_{\mathrm{F}}a_{\mathrm{f}} \lesssim 0$, we observe damped oscillation of the order parameter. We observe no atom loss on within the first 10 ms after a quench, which is two orders of magnitude longer than the times considered in the paper. We therefore exclude atom loss as a relevant factor for the observed dynamics.

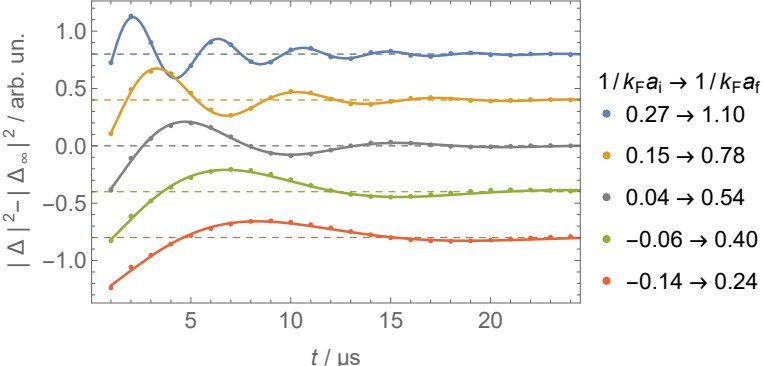

Figure 3: Decaying oscillations of the order parameter calculated in the local density approximation. Following ref. [8], we subtract the order parameter of the long-time limit $|\Delta|^2_\infty$ from our calculation and focus only on the deviations from this limit. For better visibility, different quench settings are separated by an offset as indicated by the dashed lines.

## 3 Analysis

For a quantitative analysis, we fit the measured condensate fraction with a sinusoidal function of frequency $\omega$ with exponentially damped amplitude of damping time constant $\tau$, $\sin(\omega t)\exp(-t/\tau)$. We use an exponential decay instead of the power law $t^{-1/2}$ predicted for weak excitations in the BCS model [1] because it provides more robust fitting. Additionally, we employ an empirical function to fit the background depending on the quench direction. For quenches with $1/k_F a_i < 1/k_F a_f$ this consists of two parts: Firstly, an exponential approach to a new equilibrium condensate fraction $\propto (1 - \exp(-t/\tau_2))$, which generically models the evolution of the system towards a new equilibrium state as expected in region II of the phase diagram. Secondly, an exponential decay of the condensate fraction to zero $\propto \exp(-(t - t_{\max})/\tau_3)$ starting after a variable time $t_{\max}$, which is not restricted to lie within the measured range. Such a decay is observed for many different settings and its origin – at least in part – is the inhomogeneity of the trap, as discussed below. For quenches with $1/k_F a_i > 1/k_F a_f$, the exponential approach to the new equilibrium condensate fraction is modeled as a fast decay. For longer hold times, the background function is dominated by a slower decay which may also approach finite condensate fractions. Together, this leads to the expression $c_4 \exp(-t/\tau_4) + c_5 \exp(-t/\tau_5) + c_6$.

For our theoretical analysis of the observed time series, we follow ref. [33] and use a local-density approximation. To limit complexity, we reduce the LDA approximation to its core and model the oscillatory response as a collection of independent oscillators. This approach therefore focuses on the oscillatory behaviour and neglects other effects of the quench like an expansion of the superfluid region. We note that also various other theoretical approaches and simulations have been conducted to include the effect of inhomogeneity into the frequency of the oscillation of the order parameter [9, 26, 28]. Some of these simulations highlight, in particular, the role played by the specific aspect ratios of the trap onto the non-equilibrium phase diagram [9], however, restricted to cylindrical trap geometries, which are not realized in our experiment. Within the local-density approximation, we model the oscillatory response as a collection of independent oscillators with frequencies $\tilde\omega = 2\tilde\Delta(r)/\hbar$. The gap parameter $\tilde\Delta(r)$ is set by the scattering length $a_f$ and the Fermi wave vector $k_F(r)$ determined by the local density $n(r)$ and we use the calculation of [34]. Moreover, for the quench direction $1/k_F a_i < 1/k_F a_f$ the temperature dependence of the superconducting gap $\Delta$ is included in the BCS approximation [35] using the reduced temperature $T/T_c$ from the phase diagram of the

BEC-BCS crossover [21, 22, 36], whereas for the quench direction $1/k_F a_i > 1/k_F a_f$, we employ a $T = 0$ model. Therefore, the whole cloud is then characterised by a spectrum $\mathcal{S}$ given by

$$\mathcal{S}(\omega) = \int d\tilde{\omega} \, g(\tilde{\omega}) \gamma(\omega, \tilde{\omega}) . \tag{1}$$

Here, $g(\tilde{\omega})$ is the oscillator density in frequency space at frequency $\tilde{\omega}$ and $\gamma(\omega, \tilde{\omega})$ the line shape of an oscillator with resonance frequency $\tilde{\omega}$. We calculate the oscillator density from the local value of $|\tilde{\Delta}(r)|^2$. This is motivated by the interpretation of the order parameter as the macroscopic wave function of the superfluid. The introduction of a line shape allows for the inclusion of the predicted $t^{-1/2}$ power-law decay of the Higgs oscillation [1]. We apply an inverse Fourier transform to the calculated spectra in order to compare to the dynamics we observe in our experiment. The time dynamics after the quench occurs at a non-equilibrium gap parameter, which, in general, is different from the equilibrium gap parameter set by $k_F$ and $a_f$. However, for our interaction and quench parameters, the non-equilibrium gap in a homogeneous is expected to deviate only by up to 5% from the final gap value according to [8], and we ignore this effect in our simulation for simplicity.

In Figure 3, we show the results of the LDA modelling. The numerical simulations generally show strongly-damped oscillations. However, in comparison to the experimental data, we find qualitative similarities and differences depending on the interaction range. On the BCS side and relatively close to unitarity, experiment and model both show strongly damped oscillations. For $1/k_F a_f \gtrsim 0.5$, i.e., on the BEC side, however, there is an increasing disagreement: experimentally, there is no discernible oscillation whereas the LDA model predicts an enhancement of the oscillation contrast. We therefore conclude that the intrinsic damping of the oscillations of the order parameter on the BEC side overwhelms the dephasing effect captured within the local density approximation. This could suggest one enters a new regime for the damping of the order parameter and we note that this change commences approximately where the chemical potential $\mu$ becomes negative, in agreement with [13] and qualitatively in the same range as the boundary between the II and II' phases in Figure 1. In the phase diagram of Figure 1, we have marked our measured data with labels indicating whether oscillations in the data are discernible and whether they can be explained by the local density approximation.

A feature not discussed yet is the comparably slow decay of the condensate fraction for long hold times. As evident from Figure 2a, this decay is most prominent towards the BEC limit. It is caused by a contraction of the cloud initiated by the quench in two ways. Firstly, the increased attraction strength after the quench directly leads to a contraction of the cloud [27]. Secondly, there is a change in the external potential when the magnetic field quench is performed, since the field created with the quench coil is not perfectly homogeneous. In this way, a collective mode of the trapped gas is initiated and oscillations in density and condensate fraction can be observed. Their frequency is related to the trapping frequencies and in the range of a few hundred Hz. This is two orders of magnitude slower than the Higgs oscillation and the modes can therefore be considered separately.

In order to quantitatively compare experimental and simulated data in the oscillating regime, i.e. the range $1/k_F a_f < 0.19$ for weak-to-strong quenches and $1/k_F a_f \leq 0.00$ for strong-to-weak quenches, we fit a sinusoidal oscillation with an exponentially decaying amplitude to the simulated dynamics in order to extract frequency and damping constant. In Figure 4, we compare the frequencies and damping time constants of the experimental data and the simulations. We find that the frequency of the oscillation derived from our LDA model is larger by a factor of approximately two as compared to the experiment, however, the damping rates are in very good agreement. The latter point suggests the main cause of the damping could be by the dephasing caused by the inhomogeneity of the trap [17]. The deviations of the

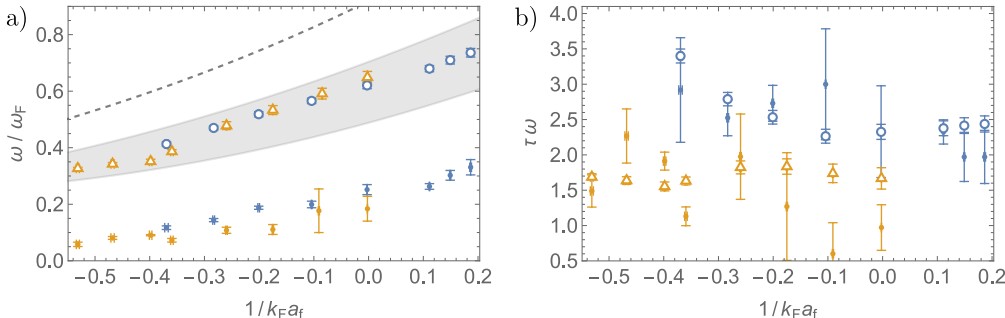

Figure 4: Oscillation frequencies and damping of the condensate fraction. a) Oscillation frequencies with full symbols showing the measurements. Blue (yellow) denotes the quench direction $1/k_{\mathrm{F}}a_{\mathrm{i}} < 1/k_{\mathrm{F}}a_{\mathrm{f}}$ ($1/k_{\mathrm{F}}a_{\mathrm{i}} > 1/k_{\mathrm{F}}a_{\mathrm{f}}$). The open symbols display the expected frequencies calculated in the local density approximation. Error bars show fit errors of the oscillatory function, and the grey-shaded band reflects the systematic uncertainty caused by approximations used in the local density approximation. The dashed line denotes the oscillation frequency derived from the calculated value of the gap at the center of the trap. b) Damping time constants. Symbol code is the same as in a). The error bars denote the uncertainty of the fitted parameters.

measured frequencies from the theoretical prediction could have contributions from several causes. (1) Our model does not take into account the effect of the interaction quench onto the gap parameter. A finite quench strength reduces the gap parameter in the final steady state, which could amount to $\sim 20\%$ for our quench parameters [7–9, 12]. In order to check the influence of the quench strength, we have performed quenches to unitarity $1/k_{\mathrm{F}}a_{\mathrm{f}} = 0$ with progressively larger values of $1/k_{\mathrm{F}}a_{\mathrm{i}} = -0.3...-0.18$, however, we found no change in the ratio $\omega/\omega_{\mathrm{F}}$. Even smaller quench amplitudes did not produce detectable signals. (2) There could be a systematic overestimation of the Fermi energy caused by deviations from the perfectly elliptic density distribution that is assumed in the inverse Abel transform. They origin of these deviations lies in the Gaussian transverse profiles of the trapping beams and the magnetic field produced by the solenoid responsible for the magnetic field quench, which produces both a magnetic offset field and a magnetic field gradient. We perform an independent check by integrating the reconstructed density profile and compare with the total atom number measured in the experiment after a short expansion of the cloud. From this we conclude that the peak density is at most a factor of two overestimated whereas a factor of four would be required in order to explain the frequency deviation. (3) The robustness of the LDA model. To this end, the quantities entering equation 1 are simplified in several ways, which also reduces the influence of some approximations. We have tested the removal of the temperature dependence of $\Delta$, assumption of a constant condensate fraction throughout the whole cloud, and have removed the intrinsic line shape. Different combinations of these contributions lead to slightly varying predictions for the oscillation frequency, which we have included as a shaded band to the theoretical prediction Figure 3a. However, a discrepancy between LDA model and experiment remains in all cases. (4) Finally, the integrable BCS model that predicts the phase diagram of Figure 1 only is an approximate solution for the strongly-interacting Fermi gas and alternative theoretical models could be considered for the explanation. For example, in the weakly-interacting BEC limit ($1/(k_F a) \gg 1$), which, however, is outside the range of our experimental parameters, Bogoliubov theory predicts a prethermalized state after an interaction quench [37]. Overall, the observed frequency remains to be explained and is an interesting starting point for further research.

# 4 Conclusion

To conclude, we have explored the quench phase diagram of a strongly-interacting superfluid Fermi gas. For a large parameter range we find the expected damped oscillations of the order parameter. The time constants for the damping are well explained by a local density approximation indicating that the observed damping could be largely due to dephasing. However, the measured frequencies deviate from the predictions of the BCS model, which requires further investigations.

## Acknowledgments

We thank K. Gao and M. Link for contributions to the initial stages of this work.

**Funding information**   This work has been supported by Deutsche Forschungsgemeinschaft through the Cluster of Excellence Matter and Light for Quantum Computing (ML4Q) EXC 2004/1–390534769 and SFB/TR 185 (project B4).

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
