# Peer review of "Quenching a Fermi superfluid across the BEC-BCS crossover"

_SciPost Physics, doi:SciPost Phys. 18, 053 (2025)_

## Round 1 · Referee Report · Anonymous (Referee 1) · 2024-9-26

Strengths

1 The considered problem of the quenches in BEC-BCS crossover in itself is quite interesting
2 The experimental results on the condensate fraction are presented with a high accuracy
3 There is a comparison between experimental and theoretical results. In some cases, there is a disagreement, which opens a question for future studies

Weaknesses

1 Presentation of the data and its interpretation can be improved

Report

The Authors consider the problem of quenching the two-component fermi gas between two values of the interaction parameter in the BCS-BEC crossover. Different physical regimes are considered and the resulting phase diagram is discussed.

Although I find the obtained results interesting, and, most probably valid, I believe that the presentation should be improved before the Manuscript can be accepted for publication.

Requested changes

1 The article uses a number of concepts to quantify the properties of the quantum system, such as superconducting parameter, order parameter, and condensate fraction. It could be useful to clearly define, what exactly is meant by each quantity.

2 The title uses the term "BEC-BCS crossover ", where the term "crossover" highlights the absence of a phase transition when changing the interaction parameter. That is, nothing special happens at the point, where the chemical potential becomes equal to zero. On the other hand, in Fig 1, seems to denote a phase transition line. If this is not the case, make it dashed and provide a discussion.

3 Figures. I find it convenient to have self-sufficient explanation of the figures in the captions. The first phrase of the caption should explain what will be shown in the figure (this is done in Fig 1, but not in Fig 2 or Fig 3). Here Fig 2 and Fig 3 are grouped, so there should be a phrase before explaining panels (a),(b) and so on. Fig 1 has regions (phases?) I, II, II', III shown but not defined in the caption. Please add a description.

4 Fig 2. The font size of the axes it much smaller as compared to the size of the text. Increase the font size. Also the panels are too small. On a computer screen one can enlarge the figures, this is not possible in a printed article. I would suggest to separate one of the panels (maybe (b)) into a different figure. Also it would be good to have a legend with the quench parameters like "$0.27 \to 1.08$" together with the color code and "BEC$\to$ BEC", "BEC $\to$ BCS", etc. title.

5 Fig2b reports "signal" on the vertical axis, change it with the quantity which is measured.

6 Figure 2 is discussed twice in the text (on page 3 and page 5). Please, introduce it better the second time. Now the discussion sounds completely disconnected. It is said that "In Figure 2, we show the results of the LDA modeling in comparison with the experimental data". Well, from the legend and the caption, it is not actually clear which data is experimental and which is the prediction.

7 There is a long text description of the piecewise function which is used but its expression is not shown. It might be useful to add it explicitly.

8 I do not understand what is meant by "two" in the following sentence "We estimate the overestimation of the density to be below two ...", please rephrase.

9 expand "BCS" abbreviation in the Abstract

10 we follow [33] change to we follow Ref. ~[33]

11 "steady-states" check hyphenation, actually it might be correct

Recommendation

Ask for minor revision

---

## Round 2 · Referee Report · Anonymous (Referee 1) · 2025-1-2

Report

The comments from my previous report were appropriately taken into account, and I find that this has clarified a number of points.

I believe that the acceptance criteria are met and I recommend publication of the article. Nevertheless, I provide an (optional) comment for Authors' consideration.

Technically speaking, the condensate fraction is the quantity that can be experimentally measured by determining the fraction of pairs at zero momentum in relation to the total number of particles. The order parameter is a theoretical concept and not directly accessible in our experiments.

Maybe it is a question of nomenclature, but within Landau's theory of phase transitions, the order parameter is a quantity that vanishes in one phase and is finite in the other phase. So, condensate fraction of molecules is the order parameter of the BEC - normal gas phase transition.

Recommendation

Publish (easily meets expectations and criteria for this Journal; among top 50%)

---

## Round 2 · Referee Report · Anonymous (Referee 2) · 2025-1-2

Strengths

1- Important new results in the field 2-Very clearly presented 3- Opens for future research directions, both in theory and experiment 4-Provides a versatile platform, in addition to superconductors, for understanding paired Fermi gases

Weaknesses

1- The alternative theory models are not described

Report

The work provides very interesting and clean experimental results concerning the dynamics of paired Fermi gases after an interaction quench. The work is novel, timely and important and it calls for further theoretical understanding, in a regime where theories are noticeably hard to develop. The work deserves publication without any hesitation.

Requested changes

1- The theory disagrees with the experiment especially in the BEC side. Is it due to approximations performed in such models? Are there other models which could be considered in such regime? Please add a discussion 2- Is it possible to explore (both theoretically and experimentally) a regime where the system behaves as point-like bosons? In this regime, what would be the predicted outcome of the quench dynamics? Please add a comment and a discussion

Recommendation

Publish (easily meets expectations and criteria for this Journal; among top 50%)

  • validity: high
  • significance: high
  • originality: top
  • clarity: top
  • formatting: perfect
  • grammar: perfect

Author:  Michael Köhl  on 2025-01-04  [id 5082]

(in reply to Report 1 on 2025-01-02)
Category:
answer to question

Dear Editor,

many thanks for sending us the referee comments on our paper. We have included the requested discussion and additional reference in our manuscript by the following paragraph:

"(4) Finally, the integrable BCS model that predicts the phase diagram of Figure 1 only is an approximate solution for the strongly-interacting Fermi gas and alternative theoretical models could be considered for the explanation. For example, in the weakly-interacting bosonic limit ($1/(k_Fa) \gg 1$), which, however, is outside the range of our experimental parameters, Bogoliubov theory predicts a prethermalized state after an interaction quench \cite{Menegoz2015}."

We have attached a revised manuscript to this response and we hope that the paper can now be accepted for publication in SciPost.

With best regards,
Michael

Attachment:

QuenchSciPost_v3.pdf

---

## Round 2 · Author Response

Dear Editor,

We thank the reviewer for her/his insightful report and the recommendation for minor revisions.
Please find our responses below. We hope that with these modifications our paper can now be accepted for publication.

With best regards,
Michael Köhl

---

## Round 2 · List of Changes

>1 The article uses a number of concepts to quantify the properties of the quantum system, such as superconducting parameter, order parameter, and condensate fraction. It could be useful to clearly define, what exactly is meant by each quantity.

Our response: The term superconducting parameter was a typo as it should have read “superconducting order parameter”. We have simplified this to order to be in line with the rest of the manuscript. We keep condensate fraction and order parameter as two different quantities. Technically speaking, the condensate fraction is the quantity that can be experimentally measured by determining the fraction of pairs at zero momentum in relation to the total number of particles. The order parameter is a theoretical concept and not directly accessible in our experiments. However, as we detail in section 2 of the manuscript, the condensate fraction serves as the proxy for the order parameter when we compare our experimental data to theoretical simulations.

>2 The title uses the term "BEC-BCS crossover ", where the term "crossover" highlights the absence of a phase transition when changing the interaction parameter. That is, nothing special happens at the point, where the chemical potential becomes equal to zero. On the other hand, in Fig 1, seems to denote a phase transition line. If this is not the case, make it dashed and provide a discussion.

Our response: The referee is correct that the BEC-BCS crossover is a crossover. This terminology refers to the behaviour of the system with respect to tuning the interaction parameter in a thermal equilibrium. It has been found, however, see, for example, reference [8], that in a non-equilibrium situation the behaviour is different. For a sudden quench taking the system our of thermal equilibrium, it has been theoretically predicted that there are different dynamical phases (labelled I,II, II’, and III) into which the system evolves, depending on the nature of the quench. These dynamical phases differ by their steady state, which, however, generally is not a thermal equilibrium state. These solid lines in Figure 1 are the phase boundaries between these dynamical phases. We have added a sentence in the caption to clarify this.

>3 Figures. I find it convenient to have self-sufficient explanation of the figures in the captions. The first phrase of the caption should explain what will be shown in the figure (this is done in Fig 1, but not in Fig 2 or Fig 3). Here Fig 2 and Fig 3 are grouped, so there should be a phrase before explaining panels (a),(b) and so on. Fig 1 has regions (phases?) I, II, II', III shown but not defined in the caption. Please add a description.

Our response: We have made the requested changes.

>4 Fig 2. The font size of the axes it much smaller as compared to the size of the text. Increase the font size. Also the panels are too small. On a computer screen one can enlarge the figures, this is not possible in a printed article. I would suggest to separate one of the panels (maybe (b)) into a different figure. Also it would be good to have a legend with the quench parameters like "0.27→1.08" together with the color code and "BEC→ BEC", "BEC → BCS", etc. title.

Our response: We have split Figure 2 into two separate Figures – one for experiment and one for simulation and we have added the labels as requested.

5 Fig2b reports "signal" on the vertical axis, change it with the quantity which is measured.

Our response: We have corrected this and now state explicitly what is plotted.

>6 Figure 2 is discussed twice in the text (on page 3 and page 5). Please, introduce it better the second time. Now the discussion sounds completely disconnected. It is said that "In Figure 2, we show the results of the LDA modeling in comparison with the experimental data". Well, from the legend and the caption, it is not actually clear which data is experimental and which is the prediction.

Our response: We have split Figure 2 into two separate Figures - see response to point 4.

>7 There is a long text description of the piecewise function which is used but its expression is not shown. It might be useful to add it explicitly.

Our response: We have rephrased the paragraph and incorporate now the fitting functions explicitly in the text.

>8 I do not understand what is meant by "two" in the following sentence "We estimate the overestimation of the density to be below two ...", please rephrase.

Our response: We have rephrased the explanation regarding point (2) and have tried to improve the clarity.

>9 expand "BCS" abbreviation in the Abstract

Our response: We have made the requested modification

>10 we follow [33] change to we follow Ref. ~[33]

Our response: We have made the modification.

>11 "steady-states" check hyphenation, actually it might be correct

Our response: We have corrected this.

---

## Round 3 · Author Response

Dear Editor,

many thanks for sending us the referee comments on our paper. We are very happy that both referees suggest our paper for publication.

Second report of Referee 1: "Maybe it is a question of nomenclature, but within Landau's theory of phase transitions, the order parameter is a quantity that vanishes in one phase and is finite in the other phase. So, condensate fraction of molecules is the order parameter of the BEC - normal gas phase transition." Our response: The referee is correct for the BEC limit. However, throughout the whole BEC-BCS crossover the consensate fraction of molecules is not a suitable choice for the order parameter but instead one chooses the superconducting gap parameter \Delta, which is theoretically defined for any interaction strength.

First report of Referee 2: "1- The theory disagrees with the experiment especially in the BEC side. Is it due to approximations performed in such models? Are there other models which could be considered in such regime? Please add a discussion 2- Is it possible to explore (both theoretically and experimentally) a regime where the system behaves as point-like bosons? In this regime, what would be the predicted outcome of the quench dynamics? Please add a comment and a discussion"

Our response: We have included the requested discussion and an additional reference in our manuscript by the following paragraph in section 3 of the manuscript: "(4) Finally, the integrable BCS model that predicts the phase diagram of Figure 1 only is an approximate solution for the strongly-interacting Fermi gas and alternative theoretical models could be considered for the explanation. For example, in the weakly-interacting BEC limit ($1/(k_Fa) \gg 1$), which, however, is outside the range of our experimental parameters, Bogoliubov theory predicts a prethermalized state after an interaction quench \cite{Menegoz2015}."

We have attached a revised manuscript to this response and we hope that the paper can now be accepted for publication in SciPost.

With best regards, Michael Köhl

---

## Editorial Decision

published